# Study protocol for the Intraoperative Complications Assessment and Reporting with Universal Standards (ICARUS) global cross-specialty surveys and consensus

**Giovanni E. Cacciamani**[1,2,3]*, **Tamir Sholklapper**[1,4], **Michael B. Eppler**[1], **Aref Sayegh**[1,5], **Lorenzo Storino Ramacciotti**[1], **Andre L. Abreu**[1,2,3], **Rene Sotelo**[1], **Mihir M. Desai**[1], **Inderbir S. Gill**[1]

**1** USC Institute of Urology and Catherine and Joseph Aresty Department of Urology, Keck School of Medicine, University of Southern California, Los Angeles, CA, United States of America, **2** Artificial Intelligence Center at USC Urology, USC Institute of Urology, University of Southern California, Los Angeles, CA, United States of America, **3** Norris Cancer Center, Keck School of Medicine, University of Southern California, Los Angeles, CA, United States of America, **4** Department of Urology, Einstein Healthcare Network, Philadelphia, PA, United States of America, **5** Department of Surgery, MedStar Good Samaritan Hospital, Baltimore, MD, United States of America

* Giovanni.cacciamani@med.usc.edu

**Data Availability Statement:** No pilot data are reported.

## Abstract

Annually, about 300 million surgeries lead to significant intraoperative adverse events (iAEs), impacting patients and surgeons. Their full extent is underestimated due to flawed assessment and reporting methods. Inconsistent adoption of new grading systems and a lack of standardization, along with litigation concerns, contribute to underreporting. Only half of relevant journals provide guidelines on reporting these events, with a lack of standards in surgical literature. To address these issues, the Intraoperative Complications Assessment and Reporting with Universal Standard (ICARUS) Global Surgical Collaboration was established in 2022. The initiative involves conducting global surveys and a Delphi consensus to understand the barriers for poor reporting of iAEs, validate shared criteria for reporting, define iAEs according to surgical procedures, evaluate the existing grading systems' reliability, and identify strategies for enhancing the collection, reporting, and management of iAEs. Invitation to participate are extended to all the surgical specialties, interventional cardiology, interventional radiology, OR Staffs and anesthesiology. This effort represents an essential step towards improved patient safety and the well-being of healthcare professionals in the surgical field.

## Introduction

Every year, approximately 300 million surgeries are performed worldwide[1, 2]. Intraoperative adverse events (iAEs) can occur, affecting patients' perioperative outcomes and survival as well as the well-being of surgeons [3, 4]. Despite their significance, the true scale of iAEs remains underestimated due to inadequate methods for assessment, collection, grading, and reporting.

**Funding:** The author(s) received no specific funding for this work.

**Competing interests:** Inderbir S. Gill has equity interest in OneLine Health and Karkinos. Mihir M. Desai is a consultant for Procept Biorobotics and Auris Surgical. Andre Luis Abreu is a consultant for Koelis and Quibim, and speaker for EDAP. Other authors do not have any competing interests. This does not alter our adherence to PLOS ONE policies on sharing data and materials

Given that these events "*do not stay in the OR*," [5] a more accurate understanding of their prevalence could aid in developing strategies to prevent them and pathways for their management [3, 6, 7].

Over the past decade, different grading systems have been introduced to capture iAEs [8, 9] However, their adoption has been limited, and the lack of consensus across specialties necessitates external assessments [8]. Additionally, the absence of a clear, standardized definition of iAEs and their effects on postoperative courses has led to inconsistency in reporting. Nowadays, recommendations for reporting adverse events are lacking in surgical literature [4, 10]. Only half of the surgery and anesthesiology journals provide guidance on reporting perioperative adverse events, and less than 0.5% recommend how to report intraoperative adverse events [11]. Not reporting these events does not mean that they didn't happen. It simply means that they were not reported, possibly due to lack of proper tools and standardized frameworks for defining, assessing, grading and collecting them [12].

Apart from the lack of standardized definitions, grading, and reporting criteria, the insufficient evaluation of these events might also be potentially associated with the effects they impose on surgeons' well-being. Preliminary findings from the pioneering BISA study [13] showed that only a handful of surgeon's report iAEs, largely out of litigation concerns and the absence of a robust reporting system. These results highlight the urgency for a global validation process to acknowledge surgeons as "second victims" and to implement support mechanisms in the aftermath of iAEs.

In 2022, the Intraoperative Complications Assessment and Reporting with Universal Standard Global Surgical Collaboration was established [10, 14] to address these issues and enhance patient safety. The ICAURS project is an global cross specialty initiative aiming at enhancing surgical research and practice by establishing and disseminating best-practice guidelines for assessing, collecting, grading and reporting iAEs across all surgical procedures. Its focus is not only on creating these guidelines but also on ensuring their practical applicability and effectiveness in clinical and academic settings. By standardizing iAE reporting, the project seeks to improve patient safety and outcomes in surgery, and it includes an evaluation component to assess the impact and refine the guidelines as needed. As part of this initiative, three global surveys and a Delphi consensus are released. These efforts aim to 1) understand the barriers for the poor reporting of iAEs and their impact on healthcare professionals, 2) validate common-shared criteria for improved iAEs reporting, 3) define iAEs according to surgical procedures and related anesthesiologic services, 4) evaluate the inter-rater reliability of existing grading systems, and 5) identify strategies to enhance the collection, reporting, and management of iAEs.

## Methods

### Selection and recruitment

The determination of the necessary sample size was made to reach a 95% confidence level and a 2% margin of error, as outlined before [15]. According to the World Healthcare Organization (WHO) Surgical Workforce Census, there were 1,853,842 surgeons and anesthesiologists around the world (https://apps.who.int/gho/data/view.main.HRSWF), thus a minimum of 2,398 respondents were calculated for the study. Corresponding authors who had published in the top-10 journals related to anesthesiology and surgery, intervention radiology and interventional cardiology between the years 2019 and 2021 were invited to participate through email. The identification of these journals and their rankings were done using the SCiMago Database (https://www.scimagojr.com). Specifically, these journals ware associated with one or more of the following specialties: Anesthesiology, Interventional Radiology, Interventional Cardiology,

**Table 1. Summary of surveys objectives, rewards, and registration number.** *Details of primary and secondary objectives are reported on ClinicalTrials.gov and in the S1 File. **Providers must complete all the rounds of the Delphi Consensus to be included in the authorship.

| Survey | Objective(s)* | Reward | IRB Number | Clinicaltrial.gov Number |
|---|---|---|---|---|
| Survey 1 | To understand the impact of iAEs on provider wellbeing. | Acknowledgment in the papers | UP-21-00473 | NCT04994392 |
| | To understand the reasons for the poor reporting of iAEs. | | | |
| | To perform a global, cross-specialty validation of the ICARUS Criteria. | | | |
| | To involve providers interested in participating in the Delphi Consensus Survey (Survey 2) | | | |
| | Developing an ecosystem for effective data collection related to iAEs | | | |
| Survey 2 | Performing a comprehensive, cross-specialty definition of Day-of-surgery AEs with a modified Delphi framework | Collaborative authorship in the papers** | | |
| Survey 3 | Evaluating the inter-rater reliability of iAEs grading systems. | Collaborative authorship in the papers | UP-21-01010 | NCT05270603 |
| | Identifying common patterns to use for proposing a new iAEs grading system | | | |

Nursing, Cardiothoracic Surgery, Colon and Rectal Surgery, General Surgery, Gynecologic Oncology, Gynecology and Obstetrics, Neurological Surgery, Ophthalmologic Surgery, Oral and Maxillofacial Surgery, Orthopaedic Surgery, Otorhinolaryngology, Plastic Surgery, Urology, Vascular Surgery. We utilized snowball sampling, encouraging respondents to share the survey with their colleagues and though social media distribution as previously done [16]. In total, 86,574 healthcare providers, consisting of 82,598 corresponding authors and 3,976 referees, were reached out to for participation in the surveys. This number does not include any additional participants from snowball sampling, or account for potential typos or defunct email addresses. To achieve a well-rounded representation encompassing diverse viewpoints, surgical specialties, and a wide spectrum of stakeholders, we will implement a purposeful over-sampling strategy. This approach guarantees the inclusion of voices from different fields, locations, and demographic groups. By taking these steps, we intend to assemble a varied and inclusive set of participants, thus boosting the reliability, relevance, and validity of our research findings. Summary of type of surveys characteristics, objectives, rewards, and registration numbers is provided in Table 1.

For each survey, baselines characteristics as detailed in Table 2 were collected.

Prior to participating in the survey, all respondents are provided with a comprehensive informed consent form detailing the purpose, procedures, potential risks, and benefits of the study; participants gave their consent by electronically acknowledging their understanding and voluntary participation.

**Table 2. Demographic domains and assessment.**

| Domains | Assessment |
|---|---|
| **Age** | Assess the respondent's age. |
| **Country** | Determine the country in which the respondent practices. |
| **Practice years** | Identify approximately how many years the respondent has been practicing. |
| **Role** | Determine the current job title or role of the respondent. |
| **Specialty** | Identify which description most closely aligns with the respondent's specialty. |
| **Surgical Approach** | Determine the surgical approach the respondent uses (or participates in) in their practice. |
| **Practice Setting** | Identify the description that best matches the respondent's practice setting. |
| **Annual Surgical Volume** | Assess approximately how many procedures or surgeries the respondent performs (or participates in) annually. |
| **Main Area of Interest** | Determine the main area of interest and daily practice of the respondent. |

Details about the aims, list of domains evaluated, and the type of analysis for each survey are reported separately as follows.

**Survey 1: Intraoperative adverse events experience, perception and reporting.** *Aims.* The purpose of this survey is three-fold. First (a), to better understand surgeons, anesthesiologists, interventional cardiologists, interventional radiologists, and nurses' perceptions and experiences surrounding iAEs event reporting. Second (b), is to globally validate the utility of a recently developed ICARUS Global Surgical collaboration Criteria [14, 17] and to determine their applicability in various surgical specialties; lastly (c) to identify strategies and tools to improve the iAEs collection and reporting. Respondents were asked the consent to be contacted for follow-up studies (survey 2 and 3).

*Survey areas, domains, and assessment.* The survey is divided in the 3 main areas (a-c) to compile with the primary study objectives as reported in Table 1.

**a) Intraoperative adverse event (iAE) experience and perception.** Overall, the section seeks to gather comprehensive information about the participants' interaction with iAEs, from their practical handling of these events to their emotional responses and beliefs about the implications of reporting such events. It also aims to understand the systemic support or challenges faced, as well as gather suggestions for improvements in the reporting process. Full list of questions is reported in the S1 File. Questions are shaped on a preliminary pilot study [10] and based on the Boston Intraoperative Adverse Events Surgeons' Attitude survey [13] and on a previews review on the potential consequences of patient's complications on surgeon wellbeing [18]. The questions in this section are designed to examine the participants' experiences, practices, beliefs, and emotions surrounding intraoperative adverse events (iAEs). The domains surveyed in this section and their assessment is reported in Table 3.

**b) Intraoperative adverse events (iAE) collection and reporting.** This section of the survey 1 aims to perform a worldwide cross-specialty assessment of the global applicability of a core set of criteria for reporting iAEs in clinical studies [10]. The focus of this assessment is to understand how universally these criteria can be applied by considering factors such as clarity, exhaustiveness, clinical usefulness, and quality assessment. The goal is to establish

**Table 3. Intraoperative adverse event (iAE) experience and perception; domains and assessment.**

| Domain | Assessment |
|---|---|
| **1. Experiences and Practices** | Frequency of witnessing iAEs in the past 12 months; Responses to iAEs; Importance of communication during or after iAEs; Debriefing and sharing practices with procedural teams, patients, and colleagues; Reporting and collecting iAEs in daily practice; Importance of regular collection of iAEs; Systems for grading iAEs. |
| **2. Practical Considerations** | Observations of iAEs; Standardized systems for assessing, reporting, and grading iAEs; Confidence in assessing and reporting iAEs; Mechanisms and support for addressing identified iAEs; Additional practical considerations impacting ability to report iAEs. |
| **3. Emotional Considerations** | Concerns about negative impacts on emotional well-being, self-confidence, and job performance; Emotional repercussions after iAEs; Additional emotional considerations related to iAE reporting. |
| **4. Perceived Benefits** | Beliefs about the clinical utility, quality assessment, enhancement of safety culture, patient safety, and educational value of reporting iAEs; Additional benefits to reporting iAEs. |
| **5. Perceived Consequences to Clinical Practice** | Potential negative impacts on risk-taking and quality of surgical practice; Additional potential consequences affecting the decision to report iAEs. |
| **6. Improvement Suggestions** | Indicators for improving or increasing iAE reporting, both in personal practice and globally; Additional comments, suggestions, questions, and concerns related to iAE reporting. |

standardized guidelines that can be adopted universally, enhancing the consistency and quality of iAE reporting worldwide [14]. The reliability and consistency between different raters are evaluated to confirm the level of agreement. To calculate the percentage of agreement, Likert scale responses are divided into two categories: scores of 4 (useful) or 5 (very useful) represent agreement, while scores of 1 (not useful), 2 (less useful), or 3 (neutral) are taken as indicative of disagreement. The validity of the criteria is assessed in 5 domains with 5-point Likert scale responses: clarity, exhaustiveness, clinical utility, quality assessment and improvement utility, and research utility [19, 20]. Full list of questions is reported in the S1 File. The domains surveyed and their assessment in this section can be grouped as reported in Table 4.

   **c) Strategies/tools to improve the iAEs collection and reporting.** This section of survey 1 goal of this section is to assess and gather feedback on various methods and tools for the assessment, grading, and reporting of intraoperative adverse events (iAEs). This includes the collection of data, standardization of reporting, efficiency of tools, and potential additional resources. Domains and their assessment are reported in Table 5.

   *Statistical analysis.* Continuous and dichotomous variables are presented as median, mean (SD), and percentages when appropriate. Inter-rater reliability and the consistency of responses are assessed to establish the level of agreement. In the determination of the percentage of agreement, Likert responses are dichotomized, with scores of 4 (useful) or 5 (very useful) indicating agreement, and scores of 1 (not useful), 2 (less useful), or 3 (neutral) indicating disagreement. Screening for outliers is performed through the evaluation of absolute individual question agreement and the distribution of responses. The internal consistency is evaluated using Cronbach's $\alpha$ [21]. For purposes of global applicability of ICARUS Global Surgical collaboration criteria (domain b) as above reported), a minimum of 80% agreement and appropriate interrater consistency (Cronbach's $\alpha > 0.5$) is required in at least 3 of the 5 domains. At least 1 of the 3 (or more) domains to achieve the minimums must include clinical utility, quality assessment, and improvement utility, and research utility. Sub-group analysis is performed for each of the surveyed specialties.

**Table 4. Intraoperative adverse Events (iAE) collection and reporting.** Domains and assessment accordingly to the ICARUS Global Surgical Collaboration Criteria [10].

| Domain | Assessment |
|---|---|
| Reporting Intraoperative Adverse Events (IAEs) | Focuses on the necessity to include IAEs as an essential outcome in perioperative study reports. |
| Definition and Reference for IAEs | IAEs and the definition of each specific IAE must be provided or referenced. |
| Classification Systems for IAEs | Each IAE should be reported using one of the proposed iAEs classification systems |
| Reporting IAEs by Grade | Focuses on reporting each IAE separately by grade. |
| Anesthesiological and Surgical Complications | Emphasizes the separate reporting of anesthesiological and surgical complications. |
| Number of IAEs and Number of Patients | Specifies the need to report the number of IAEs and the number of patients experiencing them separately. |
| Conditions Associated with IAEs | Underlines the importance of reporting conditions associated with IAEs when appropriate, following the standard criteria. |
| IAE Conversion Reporting | Discusses the necessity to report IAEs that require a conversion and the corresponding action taken, following the standard criteria. |
| Surgical Step Associated with IAEs | Calls for the reporting of the surgical step that was associated with or affected by the IAEs, in line with the usual criteria. |
| Timing of IAEs Assessment | Requires the timing of the IAEs assessment to be reported. |
| Management of IAEs | Focuses on the need to report the management of IAEs. |
| Sequelae of IAEs in Postoperative Course | Requires reporting of the sequelae of IAEs in the postoperative course, with compatibility with existing classification systems. |

**Table 5. Strategies/tools to improve the iAEs collection and reporting.** Domains and assessment.

| Domain | Assessment |
|---|---|
| **Assessment via Patient Form** | Measures the efficacy of using patient forms in capturing data for the assessment, grading, and reporting of intraoperative adverse events. This could include self-reported data or questionnaires filled out by OR staff. |
| **Utilization of Online Grade Calculator/ Converter** | Evaluates the usefulness of an online tool that calculates or converts data in order to grade and report iAEs. This could streamline data processing and standardize grading. |
| **Automated Data Recording for iAEs** | Examines the potential benefit of employing automated systems in recording data related to iAEs, which could minimize human error and enhance efficiency. |
| **Post-operative Time-out or Checklist Application** | Investigates the importance of using a checklist or time-out procedures after surgery to ensure proper data collection, and thereby assists in the accurate assessment, grading, and reporting of iAEs. |
| **Additional Resources for Data Collection** | Solicits suggestions for additional tools or resources that might be helpful in the data collection process for assessing, grading, and reporting iAEs. |
| **Scientific Publication & Criteria Checklist** | Assesses the value of having a standard criteria checklist for scientific publication of iAEs, enhancing uniformity and quality in published research. |
| **Importance of Guideline Recommendations by Academic Journals** | Evaluates how crucial it is for academic journals to offer standardized guidelines and recommendations for reporting iAEs, which can contribute to the credibility and consistency across the scientific community. |
| **Additional Resources for Scientific Publication** | Asks for extra resources or tools that might be beneficial in the process of scientifically publishing assessments, grades, and reports of iAEs, aiming to improve overall quality and consistency in academic work. |

*Survey distribution.* Google™ Forms (https://docs.google.com/forms/) is the platform used to distribute, collect, and handle the study data. The method of snowball sampling is in place to engage additional respondents. Participants must meet the following criteria to be included in the study: they must understand and willingly agree to participate; they should be proficient in English or have fluency in English medical terminology; and they must have current or previous experience with procedures or surgeries, irrespective of the specific field or domain. Responders are offered to be included in the acknowledgments section of the publications using the data retrieved, to compile with the ICJME criteria for authorship (S1 File), Table 1.

*Ethical considerations and dissemination.* This study is approved by the institutional IRB (UP-21-00473) and registered to ClinicalTrials.gov (NCT04994392). The survey outcomes will be disseminated through peer-reviewed journals, conference presentations, workshops, and webinars. A copy of the ICARUS Global surgical collaboration checklist validated though the cross specialty global survey will be uploaded in the EQUATOR Network website.

**Survey 2: Intraoperative adverse events definitions.** *Aim.* In the last two decades, the field of surgical care has grappled with the inconsistency of definitions surrounding "surgical error" and adverse events (AEs) [22, 23]. Such heterogeneity has impacted the quality of reporting and interpretation of scientific findings, reflecting the need for standardized terminology [24, 25]. Despite efforts to standardize definitions, especially pertaining to i AEs, wide acceptance of these definitions has remained elusive [26]. Current definitions for surgical complications may vary in scope and interpretation across different contexts, including preoperative, intraoperative, and postoperative stages d [27–30]. In addition, the lack of common ground in defining iAEs may contribute to underreporting [31]. Given this backdrop, the

main goal of the study is to develop a globally recognized, standardized definition of the iAEs through a Delphi consensus. This effort is not only aimed at harmonizing clinical practice and research but also at providing a comprehensive framework that encompasses various stages and aspects of surgical care, from anesthesia to postoperative events.

*Survey areas, domains, and assessment.* This international, cross-disciplinary consensus puts forward a foundational set of definitions for day-of-surgery adverse events (DOS-AEs), encompassing preoperative AEs, intraoperative AEs, and immediate postoperative stages. This set takes into consideration the timing of AE occurrence, type of surgical/interventional procedure, the specific type of AE, and the quality of AE. It includes events related to surgical or interventional procedures, anesthesiology, and nursing, and it covers both harmful and potentially harmful occurrences. In the subsequent sections, we will provide Table 6 details domains and assessment regarding each of these mentioned aspects.

*Modality.* The Delphi survey will consist of several phases, during which the panelists will evaluate and anonymously choose to agree, disagree, or propose changes to the definition essential elements. This process will be carried out through a maximum of three rounds. Following each round, participants will be given collective feedback from the prior round, assisting in the alignment of individual opinions and the formation of a consensus within the group.

Participants will assess the relevance and quality of essential elements in each in each definition for inclusion using a 1 to 5 Likert scale. A 5-point Likert scale is selected to measure consensus on each important element in the definition, with references [32, 33]. The numerical scores are defined as follows: 1: Strong agreement 2: Mild agreement 3: Indecision 4: Mild disagreement 5: Strong disagreement. In addition to utilizing the Likert scale, each question will provide a free-text space for participants to offer suggestions for improving the proposed definition or adding extra components. Participants will also have the option to select 'unable/unwilling to answer' for any of the questions.

**Table 6. Day of surgery adverse events definitions.** Domains and assessment.

| Domain | Assessment |
| --- | --- |
| **Timing** | Occurring of AEs during the preoperative, intraoperative of immediately postoperative period (24 h after surgery), specifically during anesthesia and/or the surgical/interventional procedure time is assessed. |
| | Differentiation between Anesthesia and Surgical times is tested with definitions as mentioned below. |
| | *Anesthesia Time* is a continuous time from the start of anesthesia to the end of an anesthesia service (as defined by 2019 ASA RVG). |
| | Surgical/Interventional Procedure Time is defined as the time frame from when the initial incision was made for the principal procedure to skin closure. In the case of endoscopic maneuvers which do not require any skin incision (i.e.: cystoscopy, colonoscopy, etc.) the surgical procedure time is defined as the time frame from when the endoscopic tool is initially inserted for the principal procedure through the natural or pre-existing orifice to its withdrawal.) |
| **Nature** | The AEs may be unintended, unplanned, and possibly anticipable or preventable. Differentiation between "event" and its "effect" is tested |
| **Type of Procedures Involved** | Encompasses AEs occurring during anesthesia, surgical, or interventional procedures. For iAEs, the type of surgery/intervention is also considered as Surgical Procedures Under General Anesthesia; Surgical Procedures Under Sedation; Surgical Procedures Under Local Anesthesia; Surgical Procedures without Anesthesia. |
| **Potential Harm** | AEs is potentially harmful to the patient, indicating a focus on patient safety and well-being. |
| **Recognition** | AEs may be recognized either during or after the period of interest (as reported in the domain "timing"), signifying a domain related to the detection and awareness of the event. |

*Analysis.* The agreement with definitions is being assessed on a 1 to 5-point Likert scale, and respondents who disagree, rating 1, 2, or 3, are asked to comment. Responses are analyzed, and agreement is considered achieved when 80% or more of the participants rate the criteria as 4 (agree) or 5 (strongly agree) on the Likert scale. Comments from those who disagree with the definitions are being reviewed by the study authors, and these comments are used to modify the definitions for the next round of the survey. Baseline characteristics are summarized by count and percent, and percent agreement is calculated as the proportion of those endorsing the definition with a 4 (agree) or 5 (strongly agree).

*Survey distribution.* RedCap® is the platform used to distribute, collect, and handle the study data. The method of snowball sampling is in place to engage additional respondents. Participants must meet the following criteria to be included in the study: they must understand and willingly agree to participate; they should be proficient in English or have fluency in English medical terminology; and they must have current or previous experience with procedures or surgeries, irrespective of the specific field or domain. Responders who completed all the Delphi survey rounds, are offered collaborative authorship [34] of the publications using the data retrieved under the name *ICARUS Global Surgical Collaboration Research Group*, to compile with the ICJME criteria for authorship (S1 File), Table 1.

*Ethical considerations and dissemination.* This study is approved by the institutional IRB (UP-21-00473) and registered to ClinicalTrials.gov (NCT04994392). The survey outcomes will be disseminated through peer-reviewed journals, conference presentations, workshops, and webinars.

**Survey 3: Intraoperative adverse events grading inter-rater reliability.** *Aims.* Although there are various intraoperative grading and classification systems [8], informally referred to as: EAUiaiC, iAE severity classification scheme, Modified Satava, EAES Grading system, and ClassIntra® (previously known as CLASSIC), the documentation of these iAEs is still extremely rare. Additionally, while postoperative adverse events are commonly reported, only a small portion of surgical literature addresses intraoperative complications as noteworthy outcomes [9]. The chief goal of survey 3 is to assess the uniformity and inter-rater reliability of the 5 iAE grading systems regarding the distribution of responses by quantity and percentage. Results of this survey will be instrumental for understanding the external cross-specialty variability in grading these events using the existing iAEs grading systems.

*Scenarios selection and assessment.* Each of the iAEs grading systems that has been developed currently exhibits inter-rater reliability, assessed through carefully defined surgical and anesthesiological scenarios. To maintain consistency, and with the aim of contrasting the overall inter-rater reliability with the performance specific to iAEs, we compile all the scenarios from the various iAEs grading systems papers (total 68) into an Excel spreadsheet. Subsequently, utilizing Excel's random sequence generation function, we produce a randomized selection of 10 distinct scenarios. These selected scenarios are then independently assessed and graded employing all five of the existing iAEs grading systems. We invited the respondents to elucidate their comprehension of the iAEs scenario, with the intention of assessing both their understanding of the scenario itself and the consistency between this understanding and the potential heterogeneity in the inter-rater reliability associated with the utilization of the iAEs grading systems. The questions are shaped to compile with commonly shared domains utilized in each iAEs grading systems. The domains and corresponding assessments are systematically detailed in Table 7. Details of the questions are reported in S1 File.

*Statistical analysis.* The primary data analysis requires computations to examine the uniformity and inter-rater concordance for each grading system, as detailed below:

- Distribution of responses by quantity and percentage

**Table 7. General questions for iAEs comprehension evaluation.** Domains and assessment.

| Domain | Assessment |
|---|---|
| Grading system usage | Which intraoperative adverse event grading system(s) do you use, if used? |
| Patient outcomes | Was the iAE associated with death of the patient? |
| | Was the iAE immediately life-threatening? |
| | Were there significant consequences to the patient due to the iAE? |
| Surgical errors and mishaps | Was the incorrect site, side, or surgical approach used without consent? |
| | Was the intraoperative injury missed, necessitating re-operation within 7 days of index procedure? |
| Procedural alterations and deviations | Were there any changes in the ideal intraoperative course related to iAE? |
| | Was there an unanticipated conversion of approach or significant change in planned procedure due to iAE? |
| | Was planned procedure aborted or incomplete due to iAE? |
| | Unplanned stoma as a result of iAE? |
| | Unplanned tissue or organ removal as a result of iAE? |
| | Was any surgical repair, medical treatment, or other intervention required? |
| | Was blood loss appreciably over normal range for procedure? |
| | Were 2 or more units of blood products required to manage iAE? |
| Post-operative outcomes and care | Was there a change in post-operative care due to the iAE? |
| | Did the iAE or its management necessitate intensive care admission? |

- Consistency and inter-rater reliability assessment of the 5 iAE grading systems concerning the percentage agreement of grade.

- Consistency and inter-rater reliability evaluation of the 5 iAE grading systems employing Cohen's K

- Consistency and inter-rater reliability examination of the 5 iAE grading systems utilizing the Intra-class correlation (ICC) with two-way, random effects to gauge the uniformity of grades.

- Comparison between inter-raters' reliability in grading same scenarios will be performed

- Comparison between those respondents who already utilizes one of the grading systems vs. those who don't' will be performed.

*Survey distribution.* RedCap Ⓡ is the platform used to distribute, collect, and handle the study data. The method of snowball sampling is in place to engage additional respondents. Participants must meet the following criteria to be included in the study: they must understand and willingly agree to participate; they should be proficient in English or have fluency in English medical terminology; and they must have current or previous experience with procedures or surgeries, irrespective of the specific field or domain. Responders are offered collaborative authorship [34] of the publications using the data retrieved under the name *ICARUS Global Surgical Collaboration Research Group*, to compile with the ICJME criteria for authorship (S1 File), Table 1.

*Ethical considerations and dissemination.* This study is approved by the institutional IRB (UP-21-01010) and registered to ClinicalTrials.gov (NCT05270603). The survey outcomes will be disseminated through peer-reviewed journals, conference presentations, workshops, and webinars.

## Results reporting

The results of the surveys 1,2 and 3 will be reported separately in different publications and the reporting of the survey follow the Checklist for Reporting Results of Internet E-Surveys

(CHERRIES) [35] and the American Association for public opinion Research (AAPOR) Survey Disclosure Checklist. The studies are formulated to address all applicable disclosure elements set forth by the American Association for Public Opinion Research Transparency Initiative.

## Discussion

Intraoperative adverse events are poorly reported, and their impact on both patients and surgeons is often overlooked. The goal of the ICARUS Global Surgical Collaboration project is to create an ecosystem that enhances the assessment, grading, and reporting of iAEs. This improvement aims to evaluate their impact on patients and providers, and to establish frameworks that assist surgeons in handling these effects. The project also focuses on improving patient care by implementing standardized pathways to prevent iAEs and, if they occur, to manage and follow up on them.

In the present protocol study, we delineate the objectives, scope, and methodology for a series of three global surveys. These surveys are designed to investigate the underlying causes of the inadequate reporting of specific events and to develop universally accepted definitions and criteria to bolster the collection, assessment, and reporting process. By incorporating feedback from all healthcare providers, we aim to identify effective strategies to enhance current practices. The findings from these global surveys will be instrumental in formulating widely accepted guidelines, thereby improving the assessment of these events. Consequently, the insights gained will facilitate the creation of structured frameworks, leading to the advancement of patient care.

## Supporting information

**S1 Checklist. Human participants research checklist.**
(DOCX)

**S1 File.**
(PDF)

## Author Contributions

**Conceptualization:** Giovanni E. Cacciamani, Inderbir S. Gill.

**Investigation:** Giovanni E. Cacciamani, Tamir Sholklapper, Michael B. Eppler, Aref Sayegh, Lorenzo Storino Ramacciotti, Andre L. Abreu, Rene Sotelo, Mihir M. Desai, Inderbir S. Gill.

**Methodology:** Giovanni E. Cacciamani, Inderbir S. Gill.

**Project administration:** Giovanni E. Cacciamani, Tamir Sholklapper, Michael B. Eppler, Aref Sayegh, Lorenzo Storino Ramacciotti.

**Supervision:** Giovanni E. Cacciamani, Andre L. Abreu, Rene Sotelo, Mihir M. Desai, Inderbir S. Gill.

**Visualization:** Giovanni E. Cacciamani.

**Writing – original draft:** Giovanni E. Cacciamani, Tamir Sholklapper, Michael B. Eppler, Aref Sayegh, Lorenzo Storino Ramacciotti.

**Writing – review & editing:** Giovanni E. Cacciamani, Andre L. Abreu, Rene Sotelo, Mihir M. Desai, Inderbir S. Gill.

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
