## [Decision Letter · Decision Letter 0]

23 Nov 2023

PONE-D-23-29646Study Protocol for the Intraoperative Complications Assessment and Reporting with Universal Standards (ICARUS) Global Cross-Specialty Surveys Among Surgeons, Anesthesiologists, Nurses, Interventional Cardiologists, and Interventional RadiologistsPLOS ONE

Dear Dr. Cacciamani,

Thank you for submitting your manuscript to PLOS ONE. After careful consideration, we feel that it has merit but does not fully meet PLOS ONE’s publication criteria as it currently stands. Therefore, we invite you to submit a revised version of the manuscript that addresses the points raised during the review process.

We look forward to receiving your revised manuscript.

Kind regards,

Yudai Ishiyama

Academic Editor

PLOS ONE

Journal Requirements:

"Inderbir S. Gill has equity interest in OneLine Health and Karkinos. 

Mihir M. Desai is a consultant for Procept Biorobotics and Auris Surgical.

Andre Luis Abreu is a consultant for Koelis and Quibim, and speaker for EDAP. 

Other authors do not have any competing interests."

4. One of the noted authors is a group or consortium [ICARUS Global Surgical Collaboration]. In addition to naming the author group, please list the individual authors and affiliations within this group in the acknowledgments section of your manuscript. Please also indicate clearly a lead author for this group along with a contact email address.

5. Please upload a copy of Figure 1, to which you refer in your text on page 1. If the figure is no longer to be included as part of the submission please remove all reference to it within the text.

6. Please include your tables as part of your main manuscript and remove the individual files. Please note that supplementary tables (should remain/ be uploaded) as separate "supporting information" files.

Additional Editor Comments:

This study protocol focuses on unmet clinical and research needs. Reporting intraoperative events is a complicated issue as it may directly conflict with surgeons’ benefits, as the authors mentioned.

The study protocol is neatly designed. If performed accordingly, it will contribute to improving surgical outcome research.

Please refer to the reviewers’ comments for minor issues.

Reviewers' comments:

Reviewer's Responses to Questions

**Comments to the Author**

1. Does the manuscript provide a valid rationale for the proposed study, with clearly identified and justified research questions?

Reviewer #1: Yes

Reviewer #2: Partly

2. Is the protocol technically sound and planned in a manner that will lead to a meaningful outcome and allow testing the stated hypotheses?

Reviewer #1: Yes

Reviewer #2: Partly

3. Is the methodology feasible and described in sufficient detail to allow the work to be replicable?

Reviewer #1: Yes

Reviewer #2: Yes

4. Have the authors described where all data underlying the findings will be made available when the study is complete?

Reviewer #1: Yes

Reviewer #2: Yes

5. Is the manuscript presented in an intelligible fashion and written in standard English?

Reviewer #1: Yes

Reviewer #2: Yes

6. Review Comments to the Author

You may also provide optional suggestions and comments to authors that they might find helpful in planning their study.

Reviewer #1: The paper topic is of utmost interest. This study tries to cover a lack about intraoperative adverse events report, since they have a significant impact on patients and surgeons. Despite their importance, the true scale of intraoperative adverse events remains underestimated due to inadequate methods for their assessment, collection and grading. Various grading systems have been introduced , but their adoption has been limited, with important inconsistencies in reporting. The aim of this study is to present a new method of reporting intraoperative adverse events. This study has been conducted with great scientific accuracy and the paper is well written with good English form and it's ready for its final acceptance and publication.

Reviewer #2: Major comment

Assessment methods of intraoperative complications have not been established. This study protocol could be very helpful in improving quality of daily practice. I hope that this study will contribute to the standardization of intraoperative complications assessment and reporting system and contribute to the safety of patients as well as medical staffs.

Specific comments

Title

#1 The title is a little long. Summarize the end part “Among Surgeons, Anesthesiologists, Nurses, Interventional Cardiologists, and Interventional Radiologists”.

Abstract

# It is largely redundant with the INTRODUCTION.

# With reference to previous reports, the number of surgeries performed each year is about 300 million. Please check it again.

Lancet 2015;385(Suppl 2):S11. doi:10.1016/S0140-6736(15)60806-6

BMJ 2020; 370 doi: https://doi.org/10.1136/bmj.m2917

# Delete “A sample size of 2,398 respondents was calculated for the study, with invitations extended to 86,574 healthcare providers.”. This was redundant with Materials and Methods.

Introduction

# It is largely redundant with the ABSTRACT.

# Add more specific explanation about ICARUS. In particular, mention the evaluation method.

MATERIALS AND METHODS

Selection and recruitment

# On the basis of confidence level, margin of error, and the number of surgeons from WHO’s data, sample size was determined; however, you included not only surgeons but also anesthesiologist, nurses, interventional cardiologists, and IRists in data sample. Is it reasonable.

Survey 1-3

# In these part, snowball sampling method was employed. Participants must meet the criterion on English, medical terminology in English, and current or past experience in procedure or surgeries. How was it proven that these criteria were met in this protocol? Clarify the specific criteria. Are they based only on self-reports?

DISCUSSION

# Please list some specific classifications previously reported, such as CLASSIC and ClassIntra. Additionally, mention any problems on them.

World J Surg2015;39:1663-71. doi:10.1007/s00268-015-3003-y

BMJ 2020; 370 doi: https://doi.org/10.1136/bmj.m2917

7. PLOS authors have the option to publish the peer review history of their article (what does this mean?). If published, this will include your full peer review and any attached files.

Reviewer #1: **Yes: **Antonio Luigi Pastore

Reviewer #2: No

---

## [Author Response · Author response to Decision Letter 0]

2 Jan 2024

Yudai Ishiyama

Academic Editor PLOSone 

December 26, 2023

RE: PONE-D-23-29646 Assessment and Reporting with Universal Standards (ICARUS) Global Cross-Specialty Surveys Among Surgeons, Anesthesiologists, Nurses, Interventional Cardiologists, and Interventional Radiologists

Dear Editor, 

Thank you for your review of our manuscript. We would like to thank the reviewers for their valuable time and input on this manuscript. Revisions have been made as per the reviewer recommendations. Below are the reviewers’ comments in bold font and our responses in plain font. We look forward to your review of the revised manuscript.

Dr. Giovanni Cacciamani, MD, MSC, FEBU

Associate Professor of Research Urology

Associate Professor of Research Radiology

Director Artificial Intelligence Center at USC Urology

USC Institute of Urology Catherine and Joseph Aresty Department of Urology

Keck School of Medicine, University of Southern California, Los Angeles, CA, USA.

Email: Giovanni.cacciamani@med.usc.edu

Phone: +1 (626) 4911531

Journal Requirements:

Done!

"Inderbir S. Gill has equity interest in OneLine Health and Karkinos. 

Mihir M. Desai is a consultant for Procept Biorobotics and Auris Surgical.

Andre Luis Abreu is a consultant for Koelis and Quibim, and speaker for EDAP. 

Other authors do not have any competing interests."

Done!

Yes, I confirm that the corresponding author is affiliated with the chosen institution.

4. One of the noted authors is a group or consortium [ICARUS Global Surgical Collaboration]. In addition to naming the author group, please list the individual authors and affiliations within this group in the acknowledgments section of your manuscript. Please also indicate clearly a lead author for this group along with a contact email address.

To reduce confusion we have deleted “ On behalf of ICARUS Global Surgical Collaboration”

5. Please upload a copy of Figure 1, to which you refer in your text on page 1. If the figure is no longer to be included as part of the submission please remove all reference to it within the text.

No figures are reported in the present study

6. Please include your tables as part of your main manuscript and remove the individual files. Please note that supplementary tables (should remain/ be uploaded) as separate "supporting information" files.

Done!

Done!

Done!

Additional Editor Comments:

This study protocol focuses on unmet clinical and research needs. Reporting intraoperative events is a complicated issue as it may directly conflict with surgeons’ benefits, as the authors mentioned.

The study protocol is neatly designed. If performed accordingly, it will contribute to improving surgical outcome research.

Please refer to the reviewers’ comments for minor issues.

Reply: Thank you for the feedback. We agree that the ICARUS initiative is essential for assessing the actual effects of intraoperative adverse events in surgical and anesthetic practices. By integrating diverse viewpoints, the ICARUS system aims to enhance our comprehension of intraoperative adverse events (IAEs) and develop methods to mitigate their influence on both patients and the wellbeing of surgeons.

----

Reviewer #1: 

The paper topic is of utmost interest. This study tries to cover a lack about intraoperative adverse events report, since they have a significant impact on patients and surgeons. Despite their importance, the true scale of intraoperative adverse events remains underestimated due to inadequate methods for their assessment, collection and grading. Various grading systems have been introduced , but their adoption has been limited, with important inconsistencies in reporting. The aim of this study is to present a new method of reporting intraoperative adverse events. This study has been conducted with great scientific accuracy and the paper is well written with good English form and it's ready for its final acceptance and publication.

Reply: We thank the reviewer for the comment. We believe that the ICARUS initiative is greatly needed to evaluate the real impact of intraoperative adverse events in surgery and anesthesiology. Bringing together different perspectives, the ICARUS system will help in understanding the true nature of IAEs and implementing strategies for reducing their impact on patients and surgeons' wellbeing.

-----

Reviewer #2: Major comment

Assessment methods of intraoperative complications have not been established. This study protocol could be very helpful in improving quality of daily practice. I hope that this study will contribute to the standardization of intraoperative complications assessment and reporting system and contribute to the safety of patients as well as medical staffs.

Reply: We thank the reviewer for the nice comment to our protocol.

Specific comments

Title

#1 The title is a little long. Summarize the end part “Among Surgeons, Anesthesiologists, Nurses, Interventional Cardiologists, and Interventional Radiologists”.

Reply: We agree, and we have now shortened the title as suggested in “Study Protocol for the Intraoperative Complications Assessment and Reporting with Universal Standards (ICARUS) Global Cross-Specialty Surveys and Consensus”

Abstract

# It is largely redundant with the INTRODUCTION.

Reply: the abstract is reports as per PLOS author guidelines and it report carefully the methodology used. We have edited it for better explain the rational of each survey as follow:

 ” Annually, about 300 million surgeries lead to significant intraoperative adverse events (iAEs), impacting patients and surgeons. Their full extent is underestimated due to flawed assessment and reporting methods. Inconsistent adoption of new grading systems and a lack of standardization, along with litigation concerns, contribute to underreporting. Only half of relevant journals provide guidelines on reporting these events, with a lack of standards in surgical literature. To address these issues, the Intraoperative Complications Assessment and Reporting with Universal Standard (ICARUS) Global Surgical Collaboration was established in 2022. The initiative involves conducting global surveys and a Delphi consensus to understand the barriers for poor reporting of iAEs, validate shared criteria for reporting, define iAEs according to surgical procedures, evaluate the existing grading systems' reliability, and identify strategies for enhancing the collection, reporting, and management of iAEs. Invitation to participate are extended to all the surgical specialties, interventional cardiology, interventional radiology, OR Staffs and anesthesiology. This effort represents an essential step towards improved patient safety and the well-being of healthcare professionals in the surgical field.”

Please note that the abstract complies with the PLOS author guidelines.

# With reference to previous reports, the number of surgeries performed each year is about 300 million. Please check it again. 

Lancet 2015;385(Suppl 2):S11. doi:10.1016/S0140-6736(15)60806-6

BMJ 2020; 370 doi: https://doi.org/10.1136/bmj.m2917

Reply: We thank the reviewer for the comment, and we have edited accordingly.

# Delete “A sample size of 2,398 respondents was calculated for the study, with invitations extended to 86,574 healthcare providers.”. This was redundant with Materials and Methods.

Reply: deleted as suggested.

Introduction

# It is largely redundant with the ABSTRACT.

Reply. See above. We have now edited the abstract accordingly. This is a protocol study and therefore the abstract incapsulate the rationale and the methodology used.

# Add more specific explanation about ICARUS. In particular, mention the evaluation method. 

Reply: The evaluation method is reported properly in the method, where for each survey the authors have summarized the aim, methods and analysis to be used. We added more info regarding the ICARUS Global Surgical Collaboration project as follows” The ICAURS project is an global cross specialty initiative aiming at enhancing surgical research and practice by establishing and disseminating best-practice guidelines for assessing, collecting, grading and reporting iAEs across all surgical procedures. Its focus is not only on creating these guidelines but also on ensuring their practical applicability and effectiveness in clinical and academic settings. By standardizing iAE reporting, the project seeks to improve patient safety and outcomes in surgery, and it includes an evaluation component to assess the impact and refine the guidelines as needed”.

MATERIALS AND METHODS

Selection and recruitment

# On the basis of confidence level, margin of error, and the number of surgeons from WHO’s data, sample size was determined; however, you included not only surgeons but also anesthesiologist, nurses, interventional cardiologists, and IRists in data sample. Is it reasonable.

Reply: We thank the reviewer for the comment and to judge our methodology reasonable.

Survey 1-3

# In these part, snowball sampling method was employed. Participants must meet the criterion on English, medical terminology in English, and current or past experience in procedure or surgeries. How was it proven that these criteria were met in this protocol? Clarify the specific criteria. Are they based only on self-reports?

Reply: These replies to logic questions aim to identify the profession (as indicated in tables) and language, which are self-reported. This represents a limitation that will be discussed in detail in each paper arising from the analysis of the survey results.

DISCUSSION

# Please list some specific classifications previously reported, such as CLASSIC and ClassIntra. Additionally, mention any problems on them.

Reply. We appreciate the reviewer's comment. Issues concerning each classification have been thoroughly discussed in a recent publication by our team (Sayegh, Aref S et al. “Severity Grading Systems for Intraoperative Adverse Events. A Systematic Review of the Literature and Citation Analysis.” Annals of surgery vol. 278,5 (2023): e973-e980, Reference 8 in the present study). A more in-depth discussion within the context of the present protocol for providing a reporting system (rather than a classification) would be beyond the scope of this publication and, therefore, will not be further addressed here.

---

## [Editor Report · Decision Letter 1]

15 Jan 2024

Study Protocol for the Intraoperative Complications Assessment and Reporting with Universal Standards (ICARUS) Global Cross-Specialty Surveys and Consensus

PONE-D-23-29646R1

Dear Dr. Cacciamani,

We’re pleased to inform you that your manuscript has been judged scientifically suitable for publication and will be formally accepted for publication once it meets all outstanding technical requirements.

Kind regards,

Yudai Ishiyama

Academic Editor

PLOS ONE

Additional Editor Comments (optional):

Authors provided sufficient reply to the provided comments.

---

## [Editor Report · Acceptance letter]

1 Apr 2024

PONE-D-23-29646R1 

PLOS ONE

Dear Dr. Cacciamani, 

I'm pleased to inform you that your manuscript has been deemed suitable for publication in PLOS ONE. Congratulations! Your manuscript is now being handed over to our production team.

Kind regards, 

on behalf of

Dr. Yudai Ishiyama 

Academic Editor

PLOS ONE